# ACTIVE SAMPLING FOR NODE ATTRIBUTE COMPLETION ON GRAPHS

## ABSTRACT

Node attribute is one kind of crucial information on graphs, but real-world graphs usually face attribute-missing problem where attributes of partial nodes are missing and attributes of the other nodes are available. It is meaningful to restore the missing attributes so as to benefit downstream graph learning tasks. Popular GNN is not designed for this node attribute completion issue and is not capable of solving it. Recent proposed Structure-attribute Transformer (SAT) framework decouples the input of graph structures and node attributes by a distribution matching technique, and can work on it properly. However, SAT leverages nodes with observed attributes in an equally-treated way and neglects the different contributions of different nodes in learning. In this paper, we propose a novel active sampling algorithm (ATS) to more efficiently utilize the nodes with observed attributes and better restore the missing node attributes. Specifically, ATS contains two metrics that measure the representativeness and uncertainty of each node's information by considering the graph structures, representation similarity and learning bias. Then, these two metrics are linearly combined by a Beta distribution controlled weighting scheme to finally determine which nodes are selected into the train set in the next optimization step. This ATS algorithm can be combined with SAT framework together, and is learned in an iterative manner. Through extensive experiments on 4 public benchmark datasets and two downstream tasks, we show the superiority of ATS in node attribute completion.

## 1 INTRODUCTION

Node attribute, known as a kind of important information on graphs, plays a vital role in many graph learning tasks. It boosts the performance of Graph Neural Network (GNN) Defferrard et al. (2016); Kipf & Welling (2017); Xu et al. (2019b); Veličković et al. (2018) in various domains, e.g. node classification Jin et al. (2021); Xu et al. (2019a) and community detection Sun et al. (2021); Chen et al. (2017). Meanwhile, node attribute provides human-perceptive demonstrations for the non-Euclidean structured data Zhang et al. (2019); Li et al. (2021). In spite of its indispensability, real-world graphs may have missing node attributes due to kinds of reasons Chen et al. (2022). For example, in citation graphs, key terms or detailed content of some papers may be inaccessible because of copyright protection. In social networks, profiles of some users may be unavailable due to privacy protection. When observing the attributes of partial nodes on graphs, it is significant to restore the missing attributes of the other nodes so as to benefit the downstream graph learning tasks. Namely, this is the goal of node attribute completion task.

Currently, there are limited works on the node attribute completion problem. Recent graph learning algorithms such as network embedding Cui et al. (2018) and GNN are not targeted for this problem and are limited in solving it. Random walk based methods Perozzi et al. (2014); Tang et al. (2015); Grover & Leskovec (2016) are effective in learning node embeddings on large-scale graphs. While they only take the graph structures into consideration and ignore the rich information from node attributes. Attributed random walk models Huang et al. (2019); Lei Chen & Bronstein (2019) can potentially deal with this problem but they rely on high-quality random walks and carefully designed sampling strategies which are hard to be guaranteed Yang et al. (2019). The popular GNN framework takes graph structures and node attributes as a coupled input and can work on the node attribute completion problem by some attribute-filling tricks, while these tricks introduce noise in learning and bring worse performance. In last few years, researchers begin to concentrate on the learning

problem on the attribute-missing graphs. Chen et al. (2022) propose a novel structure-attribute transformer (SAT) framework that can handle the node attribute completion case. SAT leverages structures and attributes in a decoupled scheme and achieves the joint distribution modeling by matching the latent codes of structures and attributes.

Although SAT has shown great promise on node attribute completion problem, it leverages the nodes with observed attributes in an equally-treated manner and ignores the different contributions of nodes in the learning schedule. Given limited nodes with observed attributes, it is more important to notice that different nodes have different information (e.g. degrees, neighbours, etc.) and should have different importance in the learning process. Importance re-weighting Wang et al. (2017); Fang et al. (2020); Byrd & Lipton (2019) on the optimization objective may come to mind to be a potential solution. Whereas, the information of nodes is influenced by each other and has more complex patterns. The importance distribution is implicit, intractable and rather complicated, raising great difficulties to design its formulation. It's challenging to find a more practical way to exert the different importance of the partial nodes with observed attributes at different learning stages.

In this paper, we propose an **act**ive **s**ampling algorithm named ATS to better leverage the partial nodes with observed attributes and help SAT model converge to a more desirable state. In particular, ATS measures the ***representativeness*** and ***uncertainty*** of node information on graphs to adaptively and gradually select nodes from the candidate set to the train set after each training epoch, and thus encourage the model to consider the node's importance in learning. The representativeness and uncertainty are designed by considering the graph structures, representation similarity and learning bias. Furthermore, it is interesting to find that the learning prefers nodes of high representativeness and low uncertainty at the early stage while low representativeness and high uncertainty at the late stage. Thereby, we proposes a Beta distribution controlled weighting scheme to exert adaptive learning weights on representativeness and uncertainty. In this way, these two metrics are linearly combined as the final score to determine which nodes are selected into the train set in next optimization epoch. The active sampling algorithm (ATS) and the SAT model are learned in an iterative manner until the model converges. Our contributions are as summarized follows:

- In node attribute completion, to better leverage the partial nodes with observed attributes, we advocate to use active sampling algorithm to adaptively and gradually select samples into the train set in each optimization epoch and help the model converge to a better state.

- We propose a novel ATS algorithm to measure the importance of nodes by designed ***representativeness*** and ***uncertainty*** metrics. Furthermore, when combining these two metrics as the final score function, we propose a Beta distribution controlled weighting scheme to better exert the power of ***representativeness*** and ***uncertainty*** in learning.

- We combine ATS with SAT, a newly node attribute completion model, and conduct extensive experiments on 4 public benchmarks. Through the experimental results, we show that our ATS algorithm can help SAT reach a better optimum, and restore higher-quality node attributes that benefit downstream node classification and profiling tasks.

## 2 RELATED WORK

### 2.1 DEEP GRAPH LEARNING

With the development of deep representation learning in the Euclidean vision domain Voulodimos et al. (2018), researchers have studied a lot of deep learning methods on the non-Euclidean graphs Zhang et al. (2022b). Random walk based methods can learn node embeddings by random walks , which only considers the structural information and cannot generalize to new graphs. To tackle this problem, the attributed random walk based methods (e.g.GraphRNA Huang et al. (2019)) apply random walks on both structures and attributes. These random walk based methods are practical, but they demand hardly-acquired high-quality random walks to guarantee good performance. Graph Neural Network (GNN) Scarselli et al. (2008); Defferrard et al. (2016); Kipf & Welling (2017) realizes 'graph-in, graph-out' that transforms the embeddings of node attributes while maintaining the connectivity Sanchez-Lengeling et al. (2021). GNN performs a message passing scheme, which is reminiscent of standard convolution as in Graph Convolutional Networks (GCN) Kipf & Welling (2017). GNN can infer the distribution of nodes based on node attributes and edges and achieve impressive results on graph-related tasks. There are also numerous creative modifications in GNN.

GAT Veličković et al. (2018) introduces multi-head attention into GNN. GraphSAGE Hamilton et al. (2017) moves to the inductive learning setting to deal with large-scale graphs.

Recently, more works have emphasized the importance of node attributes in graph-related downstream tasks. Both SEAL Pan et al. (2022) and WalkPool Zhang & Chen (2018) encode node representations with node attributes to achieve superior link prediction performance. In most real-world scenarios, attributes of some nodes may be inaccessible, so the node attribute completion task appears. Despite GNN's success, there are few works on this task. Recent SAT Chen et al. (2022) assumes a shared-latent space assumption on graphs and proposes a novel GNN-based distribution matching algorithm. It decouples structures and attributes and simultaneously matches the distribution of respective latent vectors. WGNN developed by Chen et al. (2021) learns node representations in Wasserstein space without any imputation. Jin et al. (2021) propose the HGNN-AC model to learn topological embedding and attribute completion with weighted aggregation. PaGNNs Jiang & Zhang (2020) can reconstruct the missing attributes based on a partial message-propagation scheme. Among them, SAT performs well and has open-source implementations, so we refer to SAT as a primary base model for completing missing node attributes.

## 2.2 ACTIVE SAMPLING ON GRAPHS

Active learning assists the model to achieve as better performance as possible while labeling as few samples as possible Ren et al. (2021). It's usually combined with deep learning model to select the most influential samples from unlabeled dataset and then label them for training to reduce the annotation cost Yoo & Kweon (2019). There are also some works of active learning on graph data. Early works Gadde et al. (2014); Gu et al. (2013); Ji & Han (2012) mainly take graph structures into consideration and design the query strategy regardless of node attributes. With the development of deep learning, many active learning algorithms are designed based on GNN. The query strategy of AGE Cai et al. (2017) measures the amount of the information contained in different nodes to select the most informative candidate node. Similar to AGE, ANRMAB Gao et al. (2018) adopts the weighted sum of different heuristics, but it adjusts the weights based on a multi-armed bandit framework. Caramalau et al. (2021) discuss two novel sampling methods: UncertainGCN and CoreGCN, which are based on uncertainty sampling and CoreSet Sener & Savarese (2017), respectively.

Nevertheless, most of today's popular active sampling algorithms on graphs aim to resolve the node classification task and focus on how to reduce the annotation cost. For this node attribute completion task, since the attribute-observed nodes are limited and the dimension of node attributes is much higher than node classes, we demand a more advanced active sampling algorithm to help the primary model utilize the attribute-observed nodes more efficiently and learn the complicated attribute distribution better. In addition, the current query strategies measure the uncertainty by an unsupervised manner, but we propose a supervised one to make the sampling closer to the primary model.

## 3 PRELIMINARY

### 3.1 PROBLEM DEFINITION

For node attribute completion task, we denote $\mathcal{G} = (\mathcal{V}, A, X)$ as a graph with node set $\mathcal{V} = \{v_1, v_2, \ldots, v_N\}$, the adjacency matrix $A \in R^{N \times N}$ and the node attribute matrix $X \in R^{N \times F}$. $\mathcal{V}^o = \{v_1^o, v_2^o, ..., v_{N_o}^o\}$ is the set of attribute-observed nodes. The attribute information of $\mathcal{V}^o$ is $X^o = \{x_1^o, x_2^o, ..., x_{N_o}^o\}$ and the structural information of $\mathcal{V}^o$ is $A^o = \{a_1^o, a_2^o, ..., a_{N_o}^o\}$. $\mathcal{V}^u = \{v_1^u, v_2^u, ..., v_{N_u}^u\}$ is the set of attribute-missing nodes. The attribute information of $\mathcal{V}^u$ is $X^u = \{x_1^u, x_2^u, ..., x_{N_u}^u\}$ and the structural information of $\mathcal{V}^u$ is $A^u = \{a_1^u, a_2^u, ..., a_{N_u}^u\}$. More specifically, we have $\mathcal{V} = \mathcal{V}^u \cup \mathcal{V}^o$, $\mathcal{V}^u \cap \mathcal{V}^o = \emptyset$, and $N = N_o + N_u$. We expect to complete the missing node attributes $X^u$ based on the observed node attributes $X^o$ and structural information $A$.

For active sampling algorithm, we denote the total training set as $T$, in which the node attributes are known. The current training set of SAT model is $T^L$ and the set containing the candidate nodes is denoted as $T^U$. We have $T = T^L \cup T^U$. We design a reasonable sampling strategy named ATS which iteratively transfers the most suitable candidate nodes from $T^U$ to $T^L$ to boost the training efficiency of SAT until $T^U = \emptyset$ and the model converges.

## 3.2 STRUCTURE-ATTRIBUTE TRANSFORMER

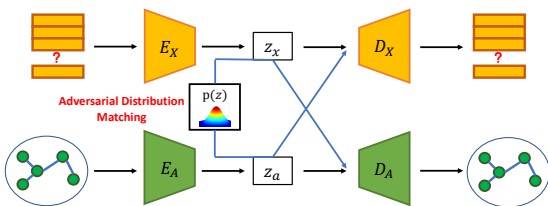

Figure 1: The general architecture of SAT. The attributes and the structure are encoded by $E_X$ and $E_A$ and reconstructed by $D_X$ and $D_A$. Meanwhile, SAT matches the latent codes of structures and attributes to a prior distribution by adversarial distribution matching.

Since we combine SAT with our ATS to demonstrate how ATS works, we briefly introduce SAT in this part. The general architecture of SAT is shown in Figure 1. SAT inputs structures and attributes in a decoupled manner, and matches the joint distribution of structures and attributes by a paired structure-attribute matching and an adversarial distribution matching. During the paired structure-attribute matching, we have a structure encoder $E_A$ (a two-layer GNN such as GCN) and an attribute encoder $E_X$ (a two-layer MLP) that encodes the structural information $a_i$ and the attribute information $x_i$ into $z_a$ and $z_x$, respectively. Then two decoders $D_A$ and $D_X$ decode $z_a$ and $z_x$ as the structures $a_i$ and attributes $x_i$ in both parallel and cross ways. Encoders and decoders are parameterized by $\phi$ and $\theta$ respectively. The joint reconstruction loss $\mathcal{L}_r$ of SAT can be written as:

$$\min_{\theta_x, \theta_a, \phi_x, \phi_a} \mathcal{L}_r = -\frac{1}{2}\mathbb{E}_{x_i}[\mathbb{E}_{q_{\phi_x}(z_x|x_i)}[\log p_{\theta_x}(x_i|z_x)]] - \frac{1}{2}\mathbb{E}_{a_i}[\mathbb{E}_{q_{\phi_a}(z_a|a_i)}[\log p_{\theta_a}(a_i|z_a)]]$$
$$- \frac{1}{2}\lambda_c \cdot \mathbb{E}_{a_i}[\mathbb{E}_{q_{\phi_a}(z_a|a_i)}[\log p_{\theta_x}(x_i|z_a)]] - \frac{1}{2}\lambda_c \cdot \mathbb{E}_{x_i}[\mathbb{E}_{q_{\phi_x}(z_x|x_i)}[\log p_{\theta_a}(a_i|z_x)]] \quad (1)$$

where $p_{\theta x}$ and $p_{\theta a}$ are the encoders, $q_{\phi x}$ and $p_{\phi a}$ are the decoders. The first two terms in Eq. 1 represent the self-reconstruction stream. The latent variable $z_x$, $z_a$ are decoded to $\hat{X}^o$, $\hat{A}$ by two-layer MLP decoders $D_X$, $D_A$ respectively. The last two terms indicate the cross-reconstruction stream, where $z_x$ and $z_a$ are decoded to $\hat{A}$ and $\hat{X}^o$, respectively.

During the adversarial distribution matching, SAT expects to match the posterior distributions $q_{\phi_x}(z_x|x_i)$ and $q_{\phi_a}(z_a|a_i)$ to a Gaussian prior $p(z) \sim \mathcal{N}(0, 1)$. Inspired by Makhzani et al. (2015), SAT adopts an efficient adversarial matching approach between $z_x$, $z_a$ and samples from the Gaussian distribution. The adversarial distribution matching loss $\mathcal{L}_{adv}$ can be written as a minimax game:

$$\min_{\psi} \max_{\phi_x, \phi_a} \mathcal{L}_{adv} = -\mathbb{E}_{z_p \sim p(z)}[\log \mathcal{D}(z_p)] - \mathbb{E}_{z_x \sim q_{\phi_x}(z_x|x_i)}[\log(1 - \mathcal{D}(z_x))]$$
$$- \mathbb{E}_{z_p \sim p(z)}[\log \mathcal{D}(z_p)] - \mathbb{E}_{z_a \sim q_{\phi_a}(z_a|a_i)}[\log(1 - \mathcal{D}(z_a))] \quad (2)$$

where $\psi$ is the parameters of the shared discriminator $\mathcal{D}$.

In summary, the objective function of SAT is:

$$\min_{\Theta} \max_{\Phi} \mathcal{L} = \mathcal{L}_r + \mathcal{L}_{adv} \quad (3)$$

where $\Theta = \{\theta_x, \theta_a, \phi_x, \phi_a, \psi\}$, $\Phi = \{\phi_x, \phi_a\}$. In the training phase of the node attribute completion task, SAT aims to minimize the reconstruction loss between $\hat{A}$, $\hat{X}^o$ and $A$, $X^o$ in Eq. 1, as well as the adversarial loss in Eq. 2. In testing, it encodes the structural information $A^u$ of attribute-missing nodes by the encoder $E_A$ and then restore their missing attributes $X^u$ by the decoder $D_X$.

## 4 METHOD

We design the query strategy of ATS by measuring the **representativeness** and **uncertainty** of the candidate nodes. Then we combine the scores of uncertainty and representativeness as the final score by an adaptive reweighting scheme and select the nodes with the highest scores for next learning epoch. We will explain these more in the following parts.

## 4.1 QUERY STRATEGY OF ATS

**Representativeness**: The major and typical patterns among the nodes are vital for the model to converge to the right direction. In this section, we introduce the concept of representativeness as a sampling metric. This metric is composed of two parts: 1) information density $\phi_{density}$ and 2) structural centrality $\phi_{centrality}$. The former mainly focuses on measuring the similarity between the corresponding latent vectors of attributes and structures. The latter indicates how closely a node is connected to its neighbours on graph. In other words, the information density is inspired by the good representation learning ability of SAT and the structural centrality is natural to mine the information on the graph structures. These two aspects offer us a comprehensive analysis of the representativeness in both implicit and explicit ways.

We first focus on the information density. SAT proposes a shared-latent space assumption for node attribute completion. We can study the node similarities through the implied features learned by the model. If there is a dense distribution of representation vectors in a local region of the latent space, the corresponding nodes will have more similar features and this region will contain further mainstream information, so we expect to train these more representative nodes in priority. For node attribute completion task, although there are attribute embeddings and structure embeddings in shared-latent space, our ATS only uses the structure embeddings $z_{a_i}$ to calculate the $\phi_{density}$ as shown in Eq. 4 since we rely on the structural representations to restore the missing node attributes. In order to find the central node located in high-density region, we employ the K-means algorithm in the latent space and calculate the Euclidean distance between each node and its clustering center. Given $d$ as the metric of Euclidean distance in $l_2$-norm and $C_{z_{a_i}}$ as the clustering center of $z_{a_i}$ in latent space, the formulation of $\phi_{density}$ is written as:

$$\phi_{density}(v_i) = \frac{1}{1 + d(z_{a_i}, C_{z_{a_i}})}, v_i \in T^U \tag{4}$$

The larger the $\phi_{density}$ is, the more representative the node is, and the node contains more representative features that are worthy of the model's attention.

Besides the feature analysis in latent space, the node representativeness can also be inferred from the explicit graph structures. We can study the connections between nodes and develop a metric to calculate the node centrality based on the structural information. Intuitively, the centrality can have a positive correlation with the number of neighbours. At the early stage of training, if we can focus on these nodes, the model will learn the approximate distribution of the data faster and reduce the influence caused by the noisy ones. PageRank Page et al. (1999) algorithm is an effective random-walk method to acquire the visiting probabilities of nodes. We utilize the PageRank score as the structural centrality $\phi_{centrality}$, which is shown as below:

$$\phi_{centrality}(v_i) = \rho \sum_j A_{ij} \frac{\phi_{centrality}(v_j)}{\sum_k A_{jk}} + \frac{1 - \rho}{N^U}, v_i \in T^U \tag{5}$$

where $N^U$ is the number of nodes in $T^U$, $\rho$ is the damping parameter. The larger $\phi_{centrality}$ is, the more representative the node is, and the node is more closely associated with its neighbours.

**Uncertainty**: Uncertainty reflects the learning state of the current model towards the nodes. When the model is reliable, it's reasonable to pay more attention on the nodes that have not been sufficiently learned. Uncertainty is a commonly-used query criterion in active learning. However, as mentioned before, the uncertainty in other sampling algorithms Cai et al. (2017); Caramalau et al. (2021); Zhang et al. (2022a) usually works for node classifications and is designed in an unsupervised manner to reduce the annotation cost. In this paper, for the node attribute completion task, in order to know the training status of the model more accurately, we consider the observed attributes and structures as supervision, and use the learning loss in SAT as the uncertainty metric, noted as $\phi_{entropy}(v_i)$.

$$\phi_{entropy}(v_i) = \mathcal{L}_r(v_i) + \mathcal{L}_{adv}(v_i), v_i \in T^U \tag{6}$$

We can input the attributes of candidate nodes and the corresponding graph structures into SAT, and then obtain their loss values. The larger $\phi_{entropy}(v_i)$ is, the more uncertainty of node $v_i$ has. From the perspective of information theory, nodes with greater uncertainty contain more information. Sampling these nodes can help the model get the information that has not been learned in previous training, thus enhancing the training efficiency.

## 4.2 SCORE FUNCTION AND BETA DISTRIBUTION CONTROLLED WEIGHTING SCHEME

We have presented three metrics of our query strategy. Then, a question arises: How to combine these metrics to score each node? Combing the metrics with a weighted sum is a possible solution but still faces great difficulties. First, the values of different metrics are incomparable because of the distinct dimensional units. Second, the different metrics may take different effects at different learning stages. To solve these issues, we introduce a percentile evaluation and design a Beta-distribution controlled re-weighting scheme to exert the functions of each metric, since Beta distribution is a suitable model for the random behavior of percentages and proportions Gupta & Nadarajah (2004).

Denote $\mathcal{P}_{\phi}(v_i, T^U)$ as the percentage of the candidate nodes in $T^U$ which have smaller values than the node $v_i$ with metric $\phi$. For example, if there are 5 candidate nodes and the scores of one metric is $[1, 2, 3, 4, 5]$, the percentile of the corresponding 5 nodes will be $[0, 0.2, 0.4, 0.6, 0.8]$. We apply the percentile to three metrics and define the final score function of ATS as:

$$S(v_i) = \alpha \cdot \mathcal{P}_{entropy}(v_i, T^U) + \beta \cdot \mathcal{P}_{density}(v_i, T^U) + \gamma \cdot \mathcal{P}_{centrality}(v_i, T^U) \qquad (7)$$

where $\alpha + \beta + \gamma = 1$. At the sampling stage, ATS will select one or several nodes with the largest $S$ and add them to the training set $T^L$ for the next training epoch of SAT.

---

**ALGORITHM 1:** ATS algorithm

---

**Input:** SAT parameters, $G$, $T^U$, $T^L$
initialization of $T^L$ and hyper-parameters;
**while** $n_e < total\_epoch$ **do**
    // Training stage
    $loss \leftarrow SAT(T^L)$;
    // Update SAT
    $loss.backward()$;
    $update(SAT.params)$;
    // Sampling stage
    **if** $\#T^U > 0$ **then**
        // SAT returns the loss values and latent representations $z_a$
        $z_a, \phi_{entropy} \leftarrow SAT(T^U)$; $\phi_{density} \leftarrow getDensity(z_a)$; $\phi_{centrality} \leftarrow getCentrality(G)$;
        $\gamma \leftarrow Beta(1, n_t)$; $\alpha, \beta \leftarrow \frac{1-\gamma}{2}$;
        $S \leftarrow \alpha \cdot \mathcal{P}_{entropy} + \beta \cdot \mathcal{P}_{density} + \gamma \cdot \mathcal{P}_{centrality}$;
        // select the node with the highest score
        $T^S \leftarrow activeSample(S, T^U)$;
        // renew the training set of SAT
        $T^L \leftarrow T^L \cup T^S$;
        // renew the candidate set
        $T^U \leftarrow T^U \setminus T^S$;
    **end**
**end**

---

Further, it is worth noting that the uncertainty and the information density are determined by the training result returned from SAT. At an early training stage, the model is unstable and the returned training result may not be quite reliable. A sampling process based on inaccurate model-returned results may lead to undesirable results. Hence, we set the weights to time-sensitive ones. The structure-related weight $\gamma$ is more credible so it can be larger initially. As the training epoch increases, the model can pay more attention to $\phi_{entropy}$ and $\phi_{density}$, while the weight $\gamma$ will decrease gradually. We formalize this by sampling $\gamma$ from a Beta distribution, of which the expectation becomes smaller with the increase of training epoch. The weighting values are defined as:

$$\gamma \sim Beta(1, n_t), \quad n_t = \frac{n_e}{\epsilon} \ and \ \alpha = \beta = \frac{1-\gamma}{2}. \qquad (8)$$

where $n_t$ is one of the determinants in Beta distribution; $\epsilon$ is used to control the expectation of $\gamma$; $n_e$ denotes the current number of epochs. We obtain the expectation by calculating the average value of 10,000 random samples.

### 4.3 ITERATIVE TRAINING AND IMPLEMENTATION

In general, our method consists of two stages: one is SAT, responsible for the training stage; the other is ATS, responsible for the sampling stage. Before the training, we divide total training set $T$ into $T^U$ and $T^L$. We randomly sample $1\%$ of the nodes in $T$ as the initial nodes of $T^L$ and the rest composes $T^U$. SAT will be trained on the changeable $T^L$. Once SAT accomplishes a single training epoch, ATS starts the sampling process. We sample the most representative and informative candidate nodes from $T^U$ according to the query strategy. These selected nodes are added to $T^L$ and removed from $T^U$. Then SAT will be trained on the renewed $T^L$ at next epoch. The training stage and the sampling stage alter iteratively until $T^U$ is null. Finally ATS is terminated and SAT will continue training to convergence. We clarify the learning process in Algorithm 1.

## 5 EXPERIMENTS AND ANALYSIS

### 5.1 DATASETS

We utilize 4 public benchmarks whose node attributes are categorical vectors. The information of used datasets is as follows: **1) Cora**. Cora McCallum et al. (2000) is a citation graph with 2,708 papers as nodes and 10,556 citation links as edges. Each node has a multi-hot attribute vector with 1,433 dimensions. The attribute vectors consist of different word tokens to determine whether they appear or not. **2) Citeseer**. Citeseer Sen et al. (2008) is another citation graph which is larger than Cora. It contains 3,327 nodes and 9,228 edges. Like Cora, each node has a multi-hot attribute vector with 3,703 dimensions. **3) Amazon-Computer** and **4) Amazon-Photo**. These two datasets are generated from Amazon co-purchase graph. The node represents the item and the edge represents the two items are usually purchased at the same time. The node attribute is a multi-hot vector with the set of words involved in the item description. Amazon-Computer Shchur et al. (2018) has 13,752 items and 245,861 edges. Amazon-Photo Shchur et al. (2018) has 7,650 nodes and 119,081 edges.

### 5.2 EXPERIMENTAL SETUP

**Baselines**: We compare SAT model combined with ATS with other baselines introduced in Chen et al. (2022): NeighAggre Şimşek & Jensen (2008), VAE Kingma & Welling (2013), GCN Kipf & Welling (2017), GraphSage Hamilton et al. (2017), GAT Veličković et al. (2018), Hers Hu et al. (2019), GraphRNA Huang et al. (2019), ARWMF Lei Chen & Bronstein (2019) and original SAT. Details about how they work on node attribute completion are illustrated in Appendix A.1.

**Evaluation metrics**: In node attribute completion, the restored attributes can provide side information for nodes and benefit downstream tasks. By following SAT Chen et al. (2022), we study the effect of ATS on two downstream tasks including node classification task in the node level and profiling task in the attribute level. In node classification, the restored attributes serve as one kind of data augmentation and supply more information to the down-stream classification task. In profiling, we aim to predict the possible profile (e.g. key terms of papers in Cora) in each attribute dimension.

**Parameters setting**: In the experiment, we randomly sample $40\%$ nodes with attributes as training data, $10\%$ nodes as validation data and the rest as test data. The attributes of validation and test nodes are unobserved in training. For the baselines, the parameters setting and the experiment results refer to Chen et al. (2022). For our ATS method, the SAT's setting remains the same, such as $\lambda_c$. We mainly have two hyper-parameters: $\epsilon$ in the weighting scheme and cluster numbers in the estimation of density $\phi_{density}$. Considering the objective of the Beta distribution weighting scheme, $\epsilon$ should be larger than the total sampling times. Hence in Cora and Citeseer, we set $\epsilon = 1500$ and when it comes to Amazon_Photo and Amazon_Computer, we set $\epsilon = 2000$. In addition, we set the cluster number as 10, 15, 10, 15 for Cora, Citeseer, Amazon_Photo and Amazon_Computer.

### 5.3 OVERALL COMPARISON

#### 5.3.1 NODE CLASSIFICATION

Classification is an effective downstream task to test the quality of the recovered attributes. In node classification task, the nodes with restored attributes are split into $80\%$ training data and $20\%$

test data. Then we conduct five-fold cross-validation in 10 times and take the average results of evaluation metrics as the model performance. We use two supervised classifiers: MLP and GCN. The MLP classifier is composed by two fully-connected layers, which classifies the nodes based on attributes. The GCN classifier is an end-to-end graph representation learning model, which can learn the structure and attributes simultaneously. Results are shown in Table 1.

According to the results of "X" row where only node attributes are used, the optimized SAT model with our proposed ATS algorithm achieves obvious improvement than original SAT model. Our ATS can also adapt to SAT with different GNN backbones (e.g. GCN and GAT) and achieve higher classification accuracy than the original models. For the results of "A+X" row where both structures and node attributes are used by a GCN classifier, our method achieves the highest score in Citeseer and Amazon-Computer, with 0.84% and 0.56% respectively, because ATS contains the density metric and can help the model better learn the inner semantic structures.

Table 1: Node classification of the node-level evaluation for node attribute completion. "X" indicates the MLP classifier that only considers the node attributes. "A+X" indicates the GCN classifier that considers both the structures and node attributes.

|  | Method | Cora | Citeseer | Amazon-Computer | Amazon-Photo |
|---|---|---|---|---|---|
| X | NeighAggre | 0.6248 | 0.5539 | 0.8365 | 0.8846 |
|  | VAE | 0.2826 | 0.2551 | 0.3747 | 0.2598 |
|  | GCN | 0.3943 | 0.3768 | 0.3660 | 0.2683 |
|  | GraphSage | 0.4852 | 0.3933 | 0.3747 | 0.2598 |
|  | GAT | 0.4143 | 0.2129 | 0.3747 | 0.2598 |
|  | Hers | 0.3046 | 0.2585 | 0.3747 | 0.2598 |
|  | GraphRNA | 0.7581 | 0.6320 | 0.6968 | 0.8407 |
|  | ARWMF | 0.7769 | 0.2267 | 0.5608 | 0.4675 |
|  | SAT(GCN) | 0.7644 | 0.6010 | 0.7410 | 0.8762 |
|  | SAT(GAT) | 0.7937 | 0.6475 | 0.8201 | 0.8976 |
|  | ATS+SAT(GCN) | 0.7850 | 0.6370 | 0.8198 | 0.8827 |
|  | ATS+SAT(GAT) | **0.8065** | **0.6662** | **0.8402** | **0.9028** |
| A+X | NeighAggre | 0.6494 | 0.5413 | 0.8715 | 0.901 |
|  | VAE | 0.3011 | 0.2663 | 0.4023 | 0.3781 |
|  | GCN | 0.4387 | 0.4079 | 0.3974 | 0.3656 |
|  | GraphSage | 0.5779 | 0.4278 | 0.4019 | 0.3784 |
|  | GAT | 0.4525 | 0.2688 | 0.4034 | 0.3789 |
|  | Hers | 0.3405 | 0.3229 | 0.4025 | 0.3794 |
|  | GraphRNA | 0.8198 | 0.6394 | 0.8650 | 0.9207 |
|  | ARWMF | 0.8025 | 0.2764 | 0.7400 | 0.6146 |
|  | SAT(GCN) | 0.8327 | 0.6599 | 0.8519 | 0.9163 |
|  | SAT(GAT) | **0.8579** | 0.6767 | 0.8766 | **0.9260** |
|  | ATS+SAT(GCN) | 0.8366 | 0.6750 | 0.8752 | 0.9181 |
|  | ATS+SAT(GAT) | 0.8573 | **0.6851** | **0.8822** | 0.9251 |

### 5.3.2 PROFILING

The model outputs the restored attributes in different dimensions with probabilities. Higher corresponding probabilities of ground-truth attributes signify better performance. In this section, we use two common metrics Recall@k and NDCG@k to evaluate the profiling performance. The experiment results are shown in Table 2.

According to the profiling results in Table 2, on the basis of the advantages established by the SAT model towards other baselines, the combination of the ATS algorithm and SAT model (ATS+SAT) obtains even higher performance in almost all the evaluation metrics and almost all the datasets. For example, ATS+SAT(GAT) obtains a relative 13.5% gain of Recall@10 and a relative 13.3% gain of NDCG@10 on Citeseer compared with SAT(GAT). The main reason of these results is that the active sampling algorithm ATS helps SAT model to realize different importance of different nodes in learning, and thus facilitates better distribution modeling of the high-dimensional node attributes.

### 5.4 STUDY OF THE WEIGHTING SCHEME

Besides the active sampling metrics, the Beta distribution controlled weighting scheme is also a highlight of the ATS algorithm. We will verify the effectiveness of our proposed scheme in comparison with other weighting schemes, such as the fixed weighting scheme and the linear variation weighting scheme. For the fixed one, the values of $\gamma$ are $0.2, \frac{1}{3}, 0.6$, and $\alpha = \beta = \frac{1-\gamma}{2}$. For the linear variation one, $\gamma$ decreases linearly from 1 to 0.5 or from 1 to 0.

Table 2: Profiling of the attribute-level evaluation for node attribute completion.

| | Method | Recall@10 | Recall@20 | Recall@50 | NDCG@10 | NDCG@20 | NDCG@50 |
|---|---|---|---|---|---|---|---|
| Cora | NeighAggre | 0.0906 | 0.1413 | 0.1961 | 0.1217 | 0.1548 | 0.1850 |
| | VAE | 0.0887 | 0.1228 | 0.2116 | 0.1224 | 0.1452 | 0.1924 |
| | GCN | 0.1271 | 0.1772 | 0.2962 | 0.1736 | 0.2076 | 0.2702 |
| | GraphSage | 0.1284 | 0.1784 | 0.2972 | 0.1768 | 0.2102 | 0.2728 |
| | GAT | 0.1350 | 0.1812 | 0.2972 | 0.1791 | 0.2099 | 0.2711 |
| | Hers | 0.1226 | 0.1723 | 0.2799 | 0.1694 | 0.2031 | 0.2596 |
| | GraphRNA | 0.1395 | 0.2043 | 0.3142 | 0.1934 | 0.2362 | 0.2938 |
| | ARWMF | 0.1291 | 0.1813 | 0.296 | 0.1824 | 0.2182 | 0.2776 |
| | SAT(GCN) | 0.1508 | 0.2182 | 0.3429 | 0.2112 | 0.2546 | 0.3212 |
| | SAT(GAT) | **0.1653** | 0.2345 | 0.3612 | 0.2250 | 0.2723 | 0.3394 |
| | ATS+SAT(GCN) | 0.1560 | 0.2259 | 0.3527 | 0.2161 | 0.2628 | 0.3298 |
| | ATS+SAT(GAT) | 0.1640 | **0.2355** | **0.3616** | **0.2258** | **0.2733** | **0.3405** |
| Citeseer | NeighAggre | 0.0511 | 0.0908 | 0.1501 | 0.0823 | 0.1155 | 0.1560 |
| | VAE | 0.0382 | 0.0668 | 0.1296 | 0.0601 | 0.0839 | 0.1251 |
| | GCN | 0.0620 | 0.1097 | 0.2052 | 0.1026 | 0.1423 | 0.2049 |
| | GraphSage | 0.0612 | 0.1097 | 0.2058 | 0.1003 | 0.1393 | 0.2034 |
| | GAT | 0.0561 | 0.1012 | 0.1957 | 0.0878 | 0.1253 | 0.1872 |
| | Hers | 0.0576 | 0.1025 | 0.1973 | 0.0904 | 0.1279 | 0.1900 |
| | GraphRNA | 0.0777 | 0.1272 | 0.2271 | 0.1291 | 0.1703 | 0.2358 |
| | ARWMF | 0.0552 | 0.1015 | 0.1952 | 0.0859 | 0.1245 | 0.1858 |
| | SAT(GCN) | 0.0764 | 0.1280 | 0.2377 | 0.1298 | 0.1729 | 0.2447 |
| | SAT(GAT) | 0.0811 | 0.1349 | 0.2431 | 0.1385 | 0.1834 | 0.2545 |
| | ATS+SAT(GCN) | 0.0854 | 0.1400 | 0.2580 | 0.1441 | 0.1896 | 0.2672 |
| | ATS+SAT(GAT) | **0.0921** | **0.1487** | **0.2635** | **0.1570** | **0.2037** | **0.2791** |
| Amazon-Computer | NeighAggre | 0.0321 | 0.0593 | 0.1306 | 0.0788 | 0.1156 | 0.1923 |
| | VAE | 0.0255 | 0.0502 | 0.1196 | 0.0632 | 0.0970 | 0.1721 |
| | GCN | 0.0273 | 0.0533 | 0.1275 | 0.0671 | 0.1027 | 0.1824 |
| | GraphSage | 0.0269 | 0.0528 | 0.1278 | 0.0664 | 0.1020 | 0.1822 |
| | GAT | 0.0271 | 0.0530 | 0.1278 | 0.0673 | 0.1028 | 0.1830 |
| | Hers | 0.0273 | 0.0525 | 0.1273 | 0.0676 | 0.1025 | 0.1825 |
| | GraphRNA | 0.0386 | 0.0690 | 0.1465 | 0.0931 | 0.1333 | 0.2155 |
| | ARWMF | 0.0280 | 0.0544 | 0.1289 | 0.0694 | 0.1053 | 0.1851 |
| | SAT(GCN) | 0.0391 | 0.0703 | 0.1514 | 0.0963 | 0.1379 | 0.2243 |
| | SAT(GAT) | 0.0421 | 0.0746 | 0.1577 | 0.1030 | 0.1463 | 0.2346 |
| | ATS+SAT(GCN) | 0.0421 | 0.0746 | 0.1575 | 0.1032 | 0.1464 | 0.2347 |
| | ATS+SAT(GAT) | **0.0440** | **0.0775** | **0.1617** | **0.1074** | **0.1519** | **0.2412** |
| Amazon-Photo | NeighAggre | 0.0329 | 0.0616 | 0.1361 | 0.0813 | 0.1196 | 0.1998 |
| | VAE | 0.0276 | 0.0538 | 0.1279 | 0.0675 | 0.1031 | 0.1830 |
| | GCN | 0.0294 | 0.0573 | 0.1324 | 0.0705 | 0.1082 | 0.1893 |
| | GraphSage | 0.0295 | 0.0562 | 0.1322 | 0.0712 | 0.1079 | 0.1896 |
| | GAT | 0.0294 | 0.0573 | 0.1324 | 0.0705 | 0.1083 | 0.1892 |
| | Hers | 0.0292 | 0.0574 | 0.1328 | 0.0714 | 0.1094 | 0.1906 |
| | GraphRNA | 0.0390 | 0.0703 | 0.1508 | 0.0959 | 0.1377 | 0.2232 |
| | ARWMF | 0.0294 | 0.0568 | 0.1327 | 0.0727 | 0.1098 | 0.1915 |
| | SAT(GCN) | 0.0410 | 0.0743 | 0.1597 | 0.1006 | 0.1450 | 0.2359 |
| | SAT(GAT) | 0.0427 | 0.0765 | 0.1635 | 0.1047 | 0.1498 | 0.2421 |
| | ATS+SAT(GCN) | 0.0426 | 0.0765 | 0.1631 | 0.1039 | 0.1491 | 0.2411 |
| | ATS+SAT(GAT) | **0.0438** | **0.0785** | **0.1651** | **0.1067** | **0.1529** | **0.2450** |

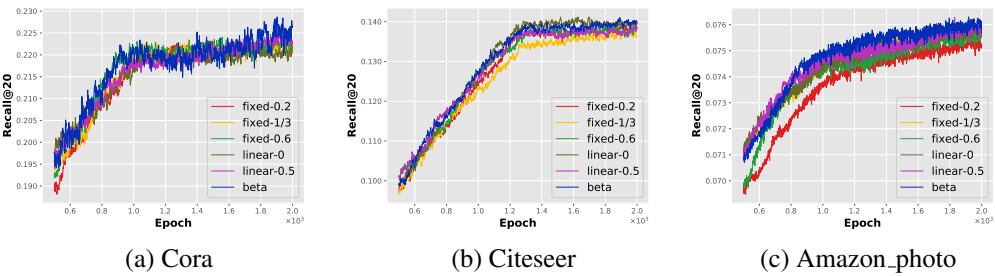

| (a) Cora | (b) Citeseer | (c) Amazon_photo |
|---|---|---|

Figure 2: Visualization of profiling performance of different weighting schemes on test data during training process. We compare our Beta distribution controlled weighting scheme with other weighting schemes(e.g. fixed weight, linear variation).

From Figure 2, we see our proposed weighting scheme outperforms other schemes because Beta distribution changes the weights dynamically during the sampling process and meanwhile remains some randomness to improve the robustness of the algorithm.

## 6 CONCLUSION

In this paper, we propose a novel active sampling algorithm ATS to better solve the node attribute completion problem. In order to distinguish the differences in the amount of information among nodes, ATS utilizes the proposed uncertainty and representativeness metrics to select the most informative nodes and renew the training set after each training epoch. Further, the Beta distribution controlled weighting scheme is proposed to adjust the metric weights dynamically according to the training status. The sampling process increases the running time of each epoch within an affordable cost, but meanwhile helps the base model achieve superior performance on profiling and node classification tasks. Therefore, ATS is effective in boosting the quality of restored attributes.

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

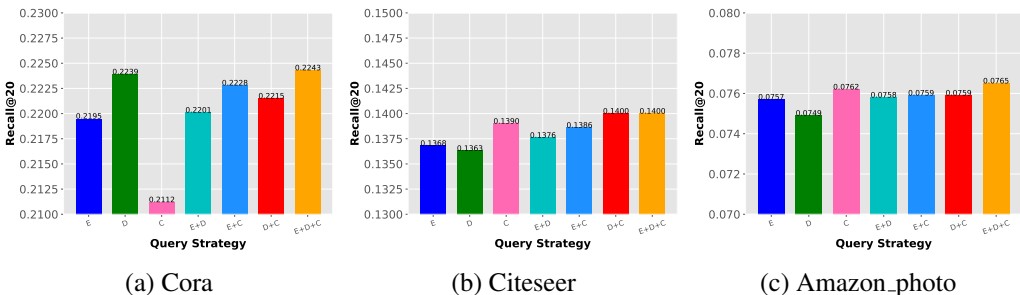

Figure 3: Ablation study of different metrics in ATS. We show the recall@20 result of different combinations of the sampling metrics on 3 benchmarks. The horizontal coordinate refers to the different sampling criteria combinations. 'E' indicates the entropy metric; 'D' indicates the density metric; 'C' indicates the centrality metric; 'E+D+C' indicates our ATS algorithm.

# A  APPENDIX

## A.1  DETAILS ABOUT THE BASELINES

NeighAggre is an intuitive attribute aggregation algorithm. It completes one node's missing attributes by averaging its neighbour nodes' attributes, which is a simple but efficient method to take advantage of the structural information. VAE is a famous generative model, which consists of an encoder and a decoder. For test nodes without the attributes, the encoder will generate the corresponding latent code through the neighbour aggregation. Then the decoder will restore the missing attributes. GCN, GraphSage and GAT are three typical graph representation learning methods. For attribute-missing scenario, only the graph structure will be encoded to latent codes. The missing attributes will be recovered by the decoders of these GNN methods from the latent code generated by the encoders. Hers is a cold-start recommendation method. GraphRNA and ARWMF are two attributed random walk based methods to learn the node representations, which can be extended to deal with the missing attributes problems. They separate the graph structure and node attributes and learn the node embeddings by random walks.

## A.2  ABLATION STUDY OF DIFFERENT METRICS IN ATS

In this section, we conduct the ablation study to investigate the effects of three different metrics in ATS. The experimental settings remain the same as the profiling task. We use Recall@20 to evaluate the performance of different metric combinations. The results are shown in Figure 3.

In Cora, centrality-only sampling method hurts the profiling performance. Different metrics focus on different aspects and the result shows that they can complement each other. The uncertainty metric focuses on the training status of the model, while the representativeness metric focuses on the implied information from both the structure and attribute aspects. Generally, any subgroup of the sampling criteria is inferior to the results achieved by the complete ATS.

## A.3  EMPIRICAL TIME COMPLEXITY ANALYSIS

Our ATS is an active sampling procedure based on the SAT model, so it's critical to study the extra processing time cost by the ATS. Thus we conduct an experiment to count the running time of different parts of the ATS compared with the original SAT. These different parts are forward process, uncertainty and representativeness. The forward process means the forward propagation, which is essential to calculate the uncertainty score. We implement the experiment on a machine with one Nvidia 1080Ti GPU.

According to the running time shown in Figure 4, the forward propagation in ATS is much faster than SAT due to the time-consuming back propagation in SAT. Although the processing time of uncertainty metric and representativeness metric is relatively higher than SAT because of the clustering and percentile calculations, it's comparable with the time of SAT. With the addition of the ATS algorithm, the time required for each epoch will increase within an acceptable range.

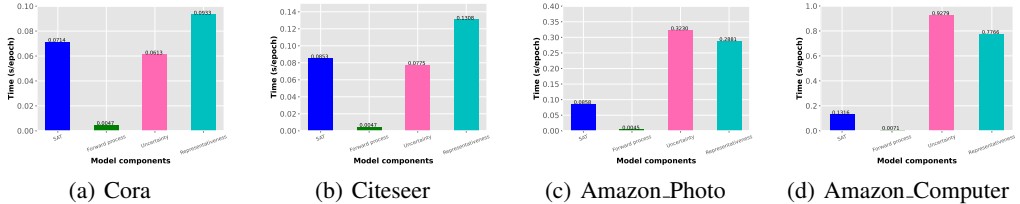

Figure 4: The comparison among the average processing GPU time per epoch of different model components. 'Forward' indicates the forward propagation that is a part of the calculation in uncertainty metric.

## A.4    SENSITIVITY OF THE HYPERPARAMETERS

As mentioned in Section 5.2, cluster number is a vital hyper-parameter that determines the information density of each node. We conduct the experiments on both the profiling and classification tasks with different cluster numbers.

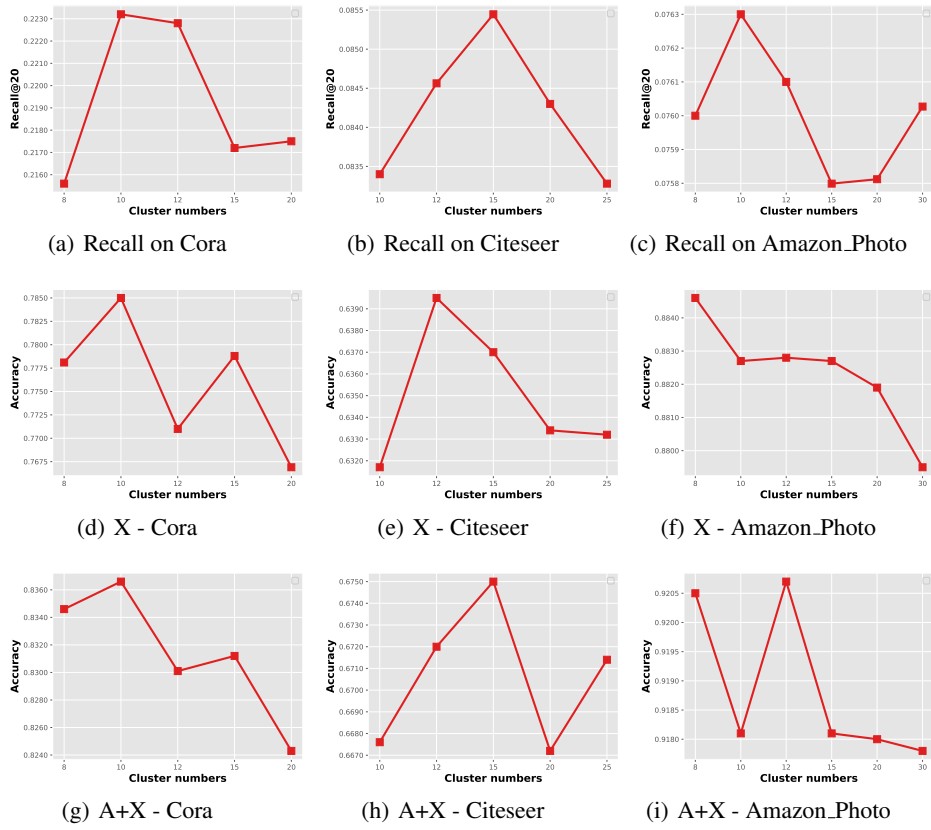

Figure 5: Results with different cluster numbers when calculating the density score in the representativeness metric. (a-c) show the Recall@20 results for profiling task. (d-f) show the attribute-only classification accuracy with the use of MLP classifier. (g-h) show the classification accuracy considering both the structure and attribute information.

The results of Figure 5 show that too large or too small cluster numbers are not conducive to the training. If there are not enough cluster centers, the sampling algorithm is not robust to extract the density of the embedding distribution. On the other hand, if there are too many cluster centers, it will

introduce more disturbance and might separate the nodes belonging to the same class. We determine the value of hyper-parameter based on the Recall@20 results in the profiling task.

