# OpenReview forum: "Active Sampling for Node Attribute Completion on Graphs"
_ICLR.cc/2023/Conference — Submitted to ICLR 2023_

### Official Review · Reviewer_TF65 · 2022-10-19

**Confidence:** 4
**Correctness:** 3
**Technical Novelty And Significance:** 2
**Empirical Novelty And Significance:** 2
**Recommendation:** 3

**Clarity, Quality, Novelty And Reproducibility:**

The technical description was clear overall. I think the introduction would benefit from some additional editing. At the moment, it reads as if it were  written in a rush.

**Strength And Weaknesses:**

Strengths:
- The criteria designed by the authors to select the next point to train on is well-motivated
- The authors leverage on a state of the art model for node attribute detection

Weaknesses:
- Lack of novelty. The main contribution is a method to prioritize training points in an active learning setting. However, the model that actually learns node attributes (which is arguably the crux of the problem) has been proposed before.
- Incremental empirical results. It is not surprising that the proposed approach marginally improves performance over SAT and GAT. Any sensible heuristic to select data points would be better than not doing anything.

**Summary Of The Paper:**

The authors propose a technique to prioritize which samples to select next in an active learning setting for node attribute completion in graphs. The proposed method uses metrics related to the "density" of a data point as well as its centrality in the graph. The proposed approach is used as input to SAT, a previously-proposed model for node attribute learning.

**Summary Of The Review:**

I think that the proposed approach is rather incremental in the literature of learning node attributes in graphs. My main criticism of the paper is that the authors rely on an existing model for the bulk of the machine learning task, and their contribution can be summarized as a heuristic method to suggest new data points in an active learning framework.

Since the work is empirical in nature, I also think that the authors could provide more evaluation on why their particular metric is of value over potential alternatives. While I agree with the authors that density and centrality are important criteria for this task, there are numerous metrics that one could come up with. For instance, instead of PageRank, we could consider node centrality or betweeness centrality.
The authors instead spend decide to report a comparison of competitors for SAT. In my opinion, this takes away from their contribution, since it makes SAT the focus of the paper. I would reframe the experiments to show how the proposed metric is superior to other density/centrality alternatives, all in the context of SAT (and potentially GAT).

Lastly, as I mentioned above, the empirical results are not surprising, since the authors are choosing points in an intelligent way. It is obvious that one should expect an improvement from this, and I don't see how the proposed work or results add much to the existing literature.

I would recommend this work to be published as a short paper, but I cannot recommend it to be featured as a full research paper.

---

> ### Author Response · Authors · 2022-11-20
> **Response to Reviewer TF65**
>
> Thank you for your comments and questions. Please find our response below.
>
> > **Lack of novelty. The main contribution is a method to prioritize training points in an active learning setting. However, the model that actually learns node attributes (which is arguably the crux of the problem) has been proposed before.**
>
> Please refer to “Response to common questions” where we have made a clarification about the target of our work the proposed model has novelty.
>
> >**I would reframe the experiments to show how the proposed metric is superior to other density/centrality alternatives, all in the context of SAT.**
>
> Please refer to “Response to common questions” where we have made a clarification about the reason why we choose PageRank as a network centrality measure.

---

### Official Review · Reviewer_RsqE · 2022-10-25

**Confidence:** 3
**Correctness:** 2
**Technical Novelty And Significance:** 1
**Empirical Novelty And Significance:** 3
**Recommendation:** 3

**Clarity, Quality, Novelty And Reproducibility:**

The paper is clearly written in some parts (e.g. the literature review and in high level description of the methodology), however, lacks clarity in the technical description. I feel that important mathematical details are missing. I am unable to comment on the originality of the work as I am not very familiar with literature on this problem.

**Strength And Weaknesses:**

Strengths: the paper tackles an important problem known to be of interest in practice (e.g. attributes of individuals are often missing due to confidentiality or privacy settings); improved performance compared to some state of the art techniques.

Weaknesses: Specifically, what is the ` node attribute completion problem' - this is clear at a high level but the complexities concerning the exact problem are never discussed (type of missing attributes, proportion of missing attributes and how it links with properties of the network). The intuition and reasoning behind some of the key choices and assumptions is not explained and remains unclear. For example, why is a shared latent space assumption required for the node attribute completion problem (second para in Section 4.1)? What do you mean by 'information density'? Is this somehow linked to network edge density? Is `structural centrality' linked to network centrality measures commonly studied in this area?
There are many vague statements here, for example (second para in Section 4.1, line 3): `If there is a dense distribution of representation vectors in a local region of the latent space, the corresponding nodes will have more similar features ..'
What does this mean exactly and why? Some mathematical notation here could have helped in stating things more precisely.
Under equation (4): 'The larger \phi_density is, the more representative the node is ....' : How does this follow from the formula above and what exactly is the node representative of and why?
Why is the PageRank score specifically used as structural centrality?


**Summary Of The Paper:**

This paper considers the problem of imputing the missing attributes of nodes in a network.This is done by identifying the most informative nodes for renewing the training set after each training epoch. Some uncertainty and representativeness metrics are proposed to achieve the task of identification. Their method achieves improved performance on node classification and profiling tasks with 4 real datasets.



**Summary Of The Review:**

The paper tackles an interesting problem and leads to improved performance, however, the assumptions and choices made for the implementation (e.g. why these are optimal/intuitive in any way) of the proposed method are not clear. Also links with existing literature on networks seems to have been ignored.

---

> ### Author Response · Authors · 2022-11-20
> **Response to Reviewer RsqE**
>
> Thank you for your comments and questions. Please find our response below.
>
> > **Why is a shared latent space assumption required for the node attribute completion problem?**
>
> The shared-latent space assumption in SAT is to better and more practically model the joint distribution $p(A, X)$ of graph structures A and node attributes X. After the joint distribution modeling, SAT can infer the missing node attributes based on $p(X|A)=p(A,X)/p(A)$. This assumption is reasonable since the structures and attributes are usually closely related to each other. In other words, SAT assumes that the attribute information and structural information of an arbitrary node have related latent variables that can generate into the attributes and structures. For more analysis about this shared-latent space assumption, please refer to the original paper[1].
>
> >**What do you mean by 'information density'? Is this somehow linked to network edge density?**
>
> Information density is a different concept compared with network edge density. Information density focuses more on the latent space, while network edge density only considers the graph structure. After encoding procedure, the latent code has more implicit information. The cluster centers in latent space signify a higher information density. If there is a dense distribution of representation vectors in a local region of the latent space, the corresponding nodes will have more similar features and this region will contain further mainstream information. Here, we try to find the cluster centers that can mostly represent the surrounding nodes in latent space.
>
> >**Is structural centrality linked to network centrality measures commonly studied in this area? Why is the PageRank score specifically used as structural centrality?**
>
> Please refer to “Response to common questions” where we have made a clarification about the reason why we choose PageRank as a network centrality measure.
>
> > **`If there is a dense distribution of representation vectors in a local region of the latent space, the corresponding nodes will have more similar features . What does this mean exactly and why?**
>
> A well-trained encoder can learn the approximate distribution of input data in latent space. In low-dimensional space, similar points should have similar features, which is a recognized principle used in many works such as AE[2], PCA[3].
>
> >**'The larger \phi_density is, the more representative the node is ....': How does this follow from the formula above and what exactly is the node representative of and why?**
>
> $\phi_{density}$ is each node’s PageRank score. PageRank is an effective random-walk method to acquire the visiting probabilities of nodes. The higher score signifies the higher visiting probabilities, which means that nodes have relatively more neighbors and then contain more structural information.
>
> [1] Xu Chen & Siheng Chen, et al (2022). Learning on attribute-missing graphs.  IEEE TPAMI.
>
> [2] Kingma, D. P., & Welling, M. (2014). Auto-encoding variational bayes.  ICLR
>
> [3] Maćkiewicz, A., & Ratajczak, W. (1993). Principal components analysis.

---

### Official Review · Reviewer_P86Z · 2022-10-28

**Confidence:** 4
**Correctness:** 3
**Technical Novelty And Significance:** 1
**Empirical Novelty And Significance:** 1
**Recommendation:** 1

**Clarity, Quality, Novelty And Reproducibility:**

Other minor issue:
Typo in Equation(2). The second z_p \sim p(z) should be z_a


**Strength And Weaknesses:**

There are some interesting points in the paper, e.g. the importance metric of the nodes. However, its overall quality is not good enough for ICLR.
1.	Motivation. First of all, the scope of the paper is too small. Node attribute completion is not a standard task, and SAT is not a well-known base model. Only designing a specific sampling method to improve the SAT model looks very trivial.
2.	I do not see the necessity of active sampling. As described in the related works, active sampling is originally used for sampling new data to label, but this paper mainly used it to adjust the training sample weights. I do not understand why they should start from a small portion of observations. They can just train a SAT using the full data, and then resample all the data to adjust the weights.
3.	Datasets. All datasets are manually made but not from a real problem (the four datasets all have full attributes). That makes the “node attribute completion” task look like a toy problem. But we know that in real world there should be such tasks. If possible real data and tasks will be much better.
4.	 Evaluation. Using the node classification task for evaluation is not reasonable. Take a very simple example, for each label we just generate a one-hot vector and use it as the missing node feature, the node classification accuracy will be 100%, but that does not mean the feature completion is good.
5.	Ablation study. The node metric has three components. I think an ablation study which removes one or two components will be necessary to show the importance of each metric component.



**Summary Of The Paper:**

This paper proposed an active sampling method to improve an existing approach called SAT for graph node attribute completion. It claims that the observed nodes are not equally important, and they designed several metrics to measure the nodes’ representativeness and uncertainty and use them for active sampling. To better control the weighting scheme, it further proposed to use Beta distribution to adjust the metric weights. Experiments show that the proposed method achieve superior performance on node classification and profiling tasks.

**Summary Of The Review:**

The solved problem is not significant, and the approach and evaluation have something unreasonable.

---

> ### Author Response · Authors · 2022-11-20
> **Response to Reviewer P86Z**
>
> Thank you for your comments and questions. Please find our response below.
>
> > **Node attribute completion is not a standard task, and SAT is not a well-known base model. Only designing a specific sampling method to improve the SAT model looks very trivial.**
>
> Node attribute completion is an emerging graph task in these three years, which has been investigated in recent works [1-3]. It is practical that graphs may have missing node attributes due to copyright protection or privacy protection in real-world scenarios. The attribute restoration can benefit several downstream tasks, which has undoubtedly an extensive application. For node attribute completion task, SAT  reaches a SOTA performance and the code is open to public.   About the novelty of this work, please refer to “Response to common questions” for response.
>
> > **They can just train a SAT using the full data, and then resample all the data to adjust the weights.**
>
> One of the objectives of our ATS is to distinguish different contributions of different nodes in learning. If we train an SAT using the full data, the resampling will become meaningless and some outlier data in full data will affect the learning procedure, which is contradictory to our motivation.
>
> > **All datasets are manually made but not from a real problem (the four datasets all have full attributes).**
>
> We have introduced four datasets in detail in Part 5.1. All the data are from the real scenarios. In addition, all the datasets should have full attributes because we need them to evaluate the generated attributes. Manual deletion of the attributes will not influence the meaning of task.
>
> > **Using the node classification task for evaluation is not reasonable.  For example, for each label you can generate a one-hot vector and use it as the missing node feature, the node classification accuracy will be 100%, but that does not mean the feature completion is good.**
>
> As is mentioned in Part 5.1, all the attributes in four datasets have real practical means (e.g. word tokens, key terms, etc.) rather than the manually-set one-hot vector as the input.
>
> >**I think an ablation study which removes one or two components will be necessary to show the importance of each metric component.**
>
> We have implemented the ablation study and the results are shown in Appendix A.2. Different metrics focus on different aspects and the result shows that they can complement each other. The uncertainty metric focuses on the training status of the model, while the representativeness metric focuses on the implied information from both the structure and attribute aspects. Generally, any subgroup of the sampling criteria is inferior to the results achieved by the complete ATS.
>
> [1] Zhixian Chen, Tengfei Ma, Yangqiu Song, and Yang Wang. Wasserstein diffusion on graphs with missing attributes. arXiv preprint arXiv:2102.03450, 2021.
>
> [2] Di Jin, Cuiying Huo, Chundong Liang, and Liang Yang. Heterogeneous graph neural network via attribute completion. In Proceedings of the Web Conference 2021, pp. 391–400, 2021.
>
> [3] Bo Jiang and Ziyan Zhang. Incomplete graph representation and learning via partial graph neural networks. arXiv preprint arXiv:2003.10130, 2020.

---

### Official Review · Reviewer_Yxf6 · 2022-10-29

**Confidence:** 4
**Correctness:** 4
**Technical Novelty And Significance:** 2
**Empirical Novelty And Significance:** Not applicable
**Recommendation:** 5

**Clarity, Quality, Novelty And Reproducibility:**

- The manuscript is straightforward to follow and the proposed method and experiments are clearly present.
- The proposed method is incremental in the sense that some common measurements are used for active sampling and it is applied to only a certain kind of method.

**Strength And Weaknesses:**

* Strengths
- The proposed node measurements are clear and reasonable to represent the importance for active sampling.
- The manuscript is easy to follow.
- The proposed method combined with SAT shows the better performance compared to baseline models

* Weakness
- The objective of the active sampling-based strategy would be great if it is clearly defined. The efficiency is described as the objective, but it needs to be further clarified and better-defined.
- Despite aiming the efficiency, the experiments or evaluations with respect to efficiency or training time are missing.
- The proposed active sampling is very tightened with SAT. It is not clear why the sampling should be coupled with SAT, not some general embedding-based learning. More generalization of the proposed method or clear explanation specific to SAT would be great to have.
- Experiments could be more enforced. The ablation study with respect to learning speed would be great to have. For example, when the same learning strategy is applied (starting from 1% training and expanding the training size over epochs), how soon does the proposed method achieve a reasonable performance as opposed to the uniform sampling? Or, how different is the learning algorithm's convergence speed when gradually expanding the training node set as opposed to directly training on the entire dataset?


**Summary Of The Paper:**

While the completion of missing node attributes is an important problem, the Structure-Attribute Transformer (SAT) is recent work that tackles that problem with leveraging all the observed attributes equally. The authors challenge the part of using all the observed attributes in an equal manner, and propose the SAT learning strategy based on active node sampling. For the active sampling, authors propose the the representativeness and the uncertain measurements for nodes on a graph and use them for active sampling. When the proposed sampling strategy is combined with SAT, it shows better performance in node classification tasks compared to the other baseline models, including the original SAT.

**Summary Of The Review:**

Despite the clearness and fair performance of the proposed method, the proposed method needs to be 1) more generalized, 2) evaluated on the matters of interests, and 3) more studied to the role of each strategy part to understand the impact of proposed methods better.

---

> ### Author Response · Authors · 2022-11-20
> **Response to Reviewer Yxf6**
>
> Thank you for your comments and questions. Please find our response below.
>
> > **The objective of the active sampling-based strategy would be great if it is clearly defined.**
>
> The objective of our active sampling algorithm is to adaptively and gradually select samples into the train set in each optimization epoch and help the model converge to a better state, which has been declared in Introduction part. In other words, compared with the original node attribute completion model, we focus more on how to help the model converge to a better optimization state and achieve better node attribute completion performance.
>
> > **Despite aiming the efficiency, the experiments or evaluations with respect to efficiency or training time are missing.**
>
> We have conducted experiments for empirical time complexity analysis. The results are shown in Appendix A.3.
> The forward propagation that calculates each node’s loss value in ATS is much faster than SAT due to the time-consuming back propagation in SAT. Although the processing time of uncertainty metric and representativeness metric is relatively higher than SAT because of the clustering and percentile calculations, it’s comparable with the time of SAT. With the addition of the ATS algorithm, the time required for each epoch will increase within an acceptable range.
>
> > **It is not clear why the sampling should be coupled with SAT, not some general embedding-based learning.**
>
> SAT is responsible for completing the missing node attributes. It’s worth mentioning that our ATS can also be combined with other attribute completion methods. Please refer to “Response to common questions” where we have made a clarification about the choice of backbone model.
>
> > **How soon does the proposed method achieve a reasonable performance as opposed to the uniform sampling?**
>
> We add a curve of uniform sampling’s profiling performance on test data to Figure2. The results on three datasets show that our method converge faster than the uniform sampling and also converge to a better state. For example in Citeseer, the original uniform sampling converge around 1800th epoch while our proposed weighting scheme converge around 1400th epoch. After 1000th epoch, the profiling performance of our weighting scheme is always better than that of uniform one.

---

### Author Response · Authors · 2022-11-20
**Response to common questions**

We thank all reviewers for their valuable feedback. We first address two common questions:
> **The novelty of the algorithm.**

ATS is proposed to more efficiently utilize the nodes with observed attributes and better restore the missing node attributes. The objective is to consider different importance of different nodes in different learning stages for the node attribute completion task. Currently, there are limited works on this.

Active learning method can help us select the most informative nodes. However, most of today’s popular active sampling algorithms on graphs aim to resolve the node classification task and focus on how to reduce the annotation cost. Unlike other existed sampling metrics, we have revised several metrics to adapt the task. For example, in density metric, we apply structure latent space to leverage the structural information. In addition, we implement a beta distribution controlled weighting scheme to exert adaptive learning weights on representativeness and uncertainty automatically.

>**ATS is highly combined with SAT?**

ATS can not only be combined with SAT, but with other models [1,2] that have training loss parts because the loss is needed for density and uncertainty scores. We can change the primary model based on different tasks. We refer to SAT as the primary model here because SAT is one SOTA model in node attribute completion and also has open-source implementations for reproducible experiments.

>**About the function of PageRank in centrality metric, why not choose other algorithms to calculate the centrality score?**

We have implemented several typical centrality algorithms (e.g. closeness centrality, betweenness centrality and PageRank centrality) to classify the nodes. We find that PageRank is the most suitable one because it reaches the highest MacroF1 score on node classification task, which can represent the centrality to the maximum extent[3].

[1] Wasserstein diffusion on graphs with missing attributes.

[2] Heterogeneous graph neural network via attribute completion.

[3] Active learning for graph embedding

---

### Decision · Program_Chairs · 2023-01-20

**Decision:**

Reject

**Justification For Why Not Higher Score:**

All agreed on Reject

**Justification For Why Not Lower Score:**

N/A

**Metareview: Summary, Strengths And Weaknesses:**

A version of a GNN is the Structure-attribute Transformer (SAT) framework which incorporates information about attributes.  Some of these attributes can be missing.  An active learning approach is used to developing a learning approach to tackle this problem.  Experiments are done on four popular datasets.  The paper is well written.

Note, the task of selecting which data to select from existing classified data is called curriculum learning.  The paper does appear to be doing curriculum learning and not active learning, as pointed out by Reviewer #P86Z (who didn't use the right term), although the methods are very related.

The results presented in Tables 1 & 2 appear good.  Reviewers would like to see additional studies, however.  One issue was the fact that the data used had complete attributes.  The test data didn't though, but for training, all attributes where complete.

Reviewers felt use of the SAT model was very specialised.  Authors point out their approach can be adapted to other GNN models.   So it should be.  As it stands the method looks incremental.

**Summary Of Ac-Reviewer Meeting:**

N/A